# Suitability of Methods to Determine Resistance to Biocidal Active Substances and Disinfectants—A Systematic Review

**Günter Kampf**

Institute for Hygiene and Environmental Medicine, University Medicine Greifswald, Ferdinand-Sauerbruch-Strasse, 17475 Greifswald, Germany; guenter.kampf@uni-greifswald.de

**Abstract:** Biocide resistance is an increasing concern. However, it is currently unknown if an elevated MIC value for an isolate correlates with a lower $\log_{10}$ reduction in suspension tests or carrier tests. The aim of this review was therefore to evaluate if isolates with a suspected tolerance to a biocidal active substance reveal an elevated MIC value and an impaired efficacy in suspension tests and carrier tests. A Medline search was done on 6 July 2022 using the following terms: "resistance biocidal MIC suspension" (16 hits), "resistance biocidal MIC carrier" (22 hits), "resistance biocidal suspension carrier" (41 hits), "tolerance biocidal MIC suspension" (1 hit), "tolerance biocidal MIC carrier" (4 hits) and "tolerance biocidal suspension carrier" (3 hits). Studies were included when a tolerance or resistance to the biocidal active substance or disinfectant was suspected and at least two of the three endpoints were evaluated in parallel in comparison to the control isolates. In three out of five studies, the elevated MIC values did not correlate with an impaired bactericidal efficacy against the biocide-tolerant isolates. In three out of five studies, an impaired activity in the suspension tests was described that correlated with an impaired efficacy in the carrier tests (peracetic acid-tolerant *K. pneumoniae* and glutaraldehyde-tolerant *M. chelonae*; the two other studies did not allow a comparison. Overall, the results from the suspension tests and tests under practical conditions allowed to determine a clinically relevant resistance.

**Keywords:** biocidal agent; antiseptic agent; disinfectant; tolerance; resistance; persistence; MIC value; suspension test; carrier test

## 1. Background

The determination of MIC values is a standard in antibiotics testing to determine susceptible and resistant isolates by measuring a range of inhibitory concentrations. Bacteria that belong to the most susceptible subpopulation and lack mechanisms of resistance are defined as susceptible isolates [1]. Resistance is the ability of bacteria to replicate and not just survive in the presence of an antibiotic. It is typically caused by inherited mutations and associated with numerous molecular mechanisms, such as the horizontal gene transfer of resistance-encoding genes encoding for antibiotic inactivating enzymes or efflux pumps [1,2]. As a result, a higher concentration of the antibiotic is required to produce the same effect in a resistant strain as is produced in a susceptible strain [3]. The MIC values are also used to describe the epidemiological cut-off (ECOFF) values, which is a measure of the antibiotic MIC distribution to separate a mostly susceptible wildtype population from a population with acquired or mutational resistance to the antibiotic. ECOFF values, however, are not the same as susceptibility breakpoints. ECOFFs have already been proposed for triclosan, benzalkonium chloride, chlorhexidine and sodium hypochlorite based on data obtained with 1635 isolates of *S. aureus*, 901 isolates of *Salmonella* spp., 368 isolates of *E. coli*, 200 isolates of *C. albicans*, 60 isolates of *K. pneumoniae*, 56 isolates of *E. faecalis*, 54 isolates of *Enterobacter* spp. and 53 isolates of *E. faecium* [4].

Suspension tests are commonly used to determine the spectrum of antimicrobial activity of disinfectants. Tests under practical conditions such as carrier tests or the four-field

test are applied to measure to antimicrobial efficacy resembling the use of the disinfectant in practice. Both types of tests measure the microbiocidal effect. The minimum requirements are defined as a $\geq 5 \log_{10}$ reduction for bacteria in the suspension tests when the disinfectant is used in human medicine [5].

It is currently not known if an elevated MIC value for a clinical or environmental isolate correlates with a lower $\log_{10}$ reduction in suspension tests or carrier tests, which could be regarded as a clinically relevant resistance to a biocidal active substance or disinfectant. The aim of this review was therefore to evaluate if isolates with a suspected tolerance to a biocidal active substance or disinfectant reveal an elevated MIC value and an impaired efficacy in suspension tests and carrier tests.

## 2. Method

A Medline search was done on 6 July 2022 using the following terms: "resistance biocidal MIC suspension" (16 hits), "resistance biocidal MIC carrier" (22 hits), "resistance biocidal suspension carrier" (41 hits), "tolerance biocidal MIC suspension" (1 hit), "tolerance biocidal MIC carrier" (4 hits) and "tolerance biocidal suspension carrier" (3 hits). The PRISMA guidelines were followed. All studies were screened for original data on the MIC values, as well as $\log_{10}$ reductions obtained in suspension tests and in carrier tests. All studies were included in which the authors described isolates with a tolerance or resistance to the biocidal active substance or disinfectant, and at least two of the three endpoints were evaluated in parallel in comparison to the control isolates. Only articles in English were included. Studies were excluded when only one of the three endpoints was described, or no tolerance or resistance was suspected by the authors. Reviews were not included but screened for any information relevant to the scope of this review. Two additional studies fulfilling the inclusion criteria but which were not found by the literature search were added from a personal library (Figure 1).

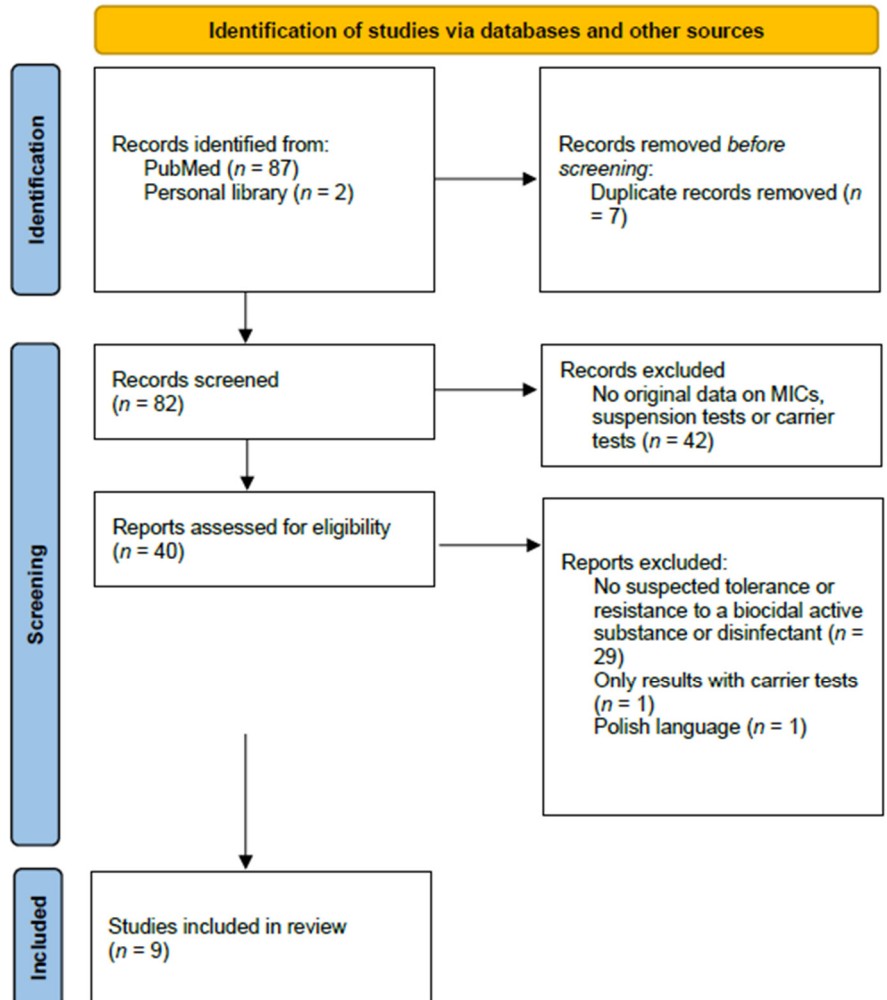

**Figure 1.** Flow diagram on the study selection, exclusion and inclusion within the systematic review (PRISMA).

### 3. Results

*3.1. Can Elevated MIC Values Predict an Impaired Efficacy in Suspension Tests?*

Five studies were found with comparative susceptibility data based on MIC values and efficacy data for biocidal active substances or disinfectants measured in suspension tests (Tables 1 and 2).

**Table 1.** Overview on the MIC values and mean $\log_{10}$ reductions from suspension tests obtained with standard bacterial test strains and isolates with a suspected tolerance to biocidal active substances or disinfectants.

| Species | Biocidal Active Substance | Strain | MIC (mg/L) | Mean Log$_{10}$ Reduction | Exposure Time | Reference |
|---|---|---|---|---|---|---|
| *Providencia* spp. | 0.01% CHG * | Four strains (suspected tolerance to CHG) | ≥400 ** | 0.5–1.1 | 10 min | [6] |
| | | Two control strains ("susceptible to CHG") | No data | 4.8–5.1 | 10 min | [6] |
| | 0.01% BAC * | Four strains (suspected tolerance to CHG) | ≥400 ** | 0.5–1.2 | 10 min | [6] |
| | | Two control strains ("susceptible to CHG") | No data | 5.7–6.3 | 10 min | [6] |

Table 1. *Cont.*

| Species | Biocidal Active Substance | Strain | MIC (mg/L) | Mean $Log_{10}$ Reduction | Exposure Time | Reference |
|---|---|---|---|---|---|---|
| *Burkholderia cepacia* complex | 4% CHG * | ATCC 17616 | 10–20 ** | $\geq$5.0 | 5 min | [7] |
| | | LMG 18821 | 90–100 ** | $\geq$5.0 | 5 min | [7] |
| | | LMG 16660 | 90–100 ** | $\geq$5.0 | 5 min | [7] |
| | | J 2315 | >100 ** | <5.0 | 1 h | [7] |
| | | Novel group K strain 24 | 90–100 ** | $\geq$5.0 | 5 min | [7] |
| *Acinetobacter* spp. | 0.0032% CHG * | 20 non-repetitive clinical isolates | 10 ** | 5.0 | 5 min | [8] |
| | | 28 non-repetitive clinical isolates | >50 ** | 5.0 *** | 5 min | [8] |
| | 0.0032% BAC * | 20 non-repetitive clinical isolates | 10 ** | 5.0 | 5 min | [8] |
| | | 28 non-repetitive clinical isolates | >50 ** | 5.0 *** | 5 min | [8] |
| | 0.0064% alkyldiaminoethyl-glycine hydrochloride * | 20 non-repetitive clinical isolates | 10–50 ** | 5.0 | 5 min | [8] |
| | | 28 non-repetitive clinical isolates | >100 ** | 5.0 *** | 5 min | [8] |
| *S. aureus* | 0.01% triclosan * | ATCC 6538 | 0.12 **** | 0.3 | 5 min | [9] |
| | | QBR-102278-1177 | 4 **** | 0.2 | 5 min | [9] |
| | | QBR-102278-1219 | 4 **** | 0.3 | 5 min | [9] |
| | | QBR-102278-1619 | 4 **** | 0.4 | 5 min | [9] |
| *S. aureus* | 0.06% triclosan * | ATCC 6538 | 0.12 **** | 5.5 | 5 min | [9] |
| | | QBR-102278-1177 | 4 **** | 4.0 | 5 min | [9] |
| | | QBR-102278-1219 | 4 **** | 4.0 | 5 min | [9] |
| | | QBR-102278-1619 | 4 **** | 4.7 | 5 min | [9] |
| *S. aureus* | 0.1% triclosan * | ATCC 6538 | 0.12 **** | >5.5 | 5 min | [9] |
| | | QBR-102278-1177 | 4 **** | 5.5 | 5 min | [9] |
| | | QBR-102278-1219 | 4 **** | 4.0 | 5 min | [9] |
| | | QBR-102278-1619 | 4 **** | 5.5 | 5 min | [9] |

* Solution of biocidal active substance; ** CHG MIC value; *** strains with higher MIC values showed significantly lower $log_{10}$ reductions in 5 min at two-fold and four-fold lower concentrations of the biocidal active substance; **** triclosan MIC value.

**Table 2.** Overview on MIC values, mean $log_{10}$ reductions from suspension tests and carrier tests obtained with standard bacterial test strains and isolates with a suspected tolerance to chlorhexidine diacetate.

| Species | Biocidal Active Substance | Strain | MIC Value (mg/L) | Suspension Test | | Carrier Test | | Reference |
|---|---|---|---|---|---|---|---|---|
| | | | | Mean $log_{10}$ reduction | Exposure time | Mean $log_{10}$ reduction | Exposure time | |
| *P. aeruginosa* | 0.01% chlorhexi-dine diacetate * | NCIMB 10421 | 8–10 ** | 3.8 | 5 min | 1.5 | 5 min | [10] |
| | | Strain Pa 6 | 28 ** | 4.0 | 5 min | 2.2 | 5 min | [10] |
| | | Strain Pa 7 | >40 ** | 3.8 | 5 min | 1.4 | 5 min | [10] |
| | | Strain Pa 8 | >50 ** | 4.1 | 5 min | 1.2 | 5 min | [10] |
| | | Strain Pa 9 | 70 ** | 4.2 | 5 min | 2.0 | 5 min | [10] |
| | | Strain Pa 51a | >70 ** | 4.0 | 5 min | 1.5 | 5 min | [10] |
| | | Strain Pa 54a | >70 ** | 3.9 | 5 min | 1.8 | 5 min | [10] |

* Solution of biocidal active substance; ** chlorhexidine diacetate MIC value.

Stickler et al. compared in 1976 the susceptibility of *Providencia* spp. strains to CHG and determined if strains with a high MIC value are more difficult to kill in suspension tests

(Table 1). Four strains were considered to be resistant to CHG with MIC values $\geq$ 400 mg/L and revealed a $\log_{10}$ reduction by exposure to 100 mg/L CHG for 10 min between 0.5 and 1.1, whereas the two susceptible control strains were killed more effectively (4.8–5.1 $\log_{10}$). BAC (100 mg/L) was also less effective against the strains with high CHG MIC values (0.5–1.2 $\log_{10}$) compared to the two control strains (5.7–6.3 $\log_{10}$), suggesting a cross-resistance. A similar susceptibility of the two groups of strains, however, was found to povidone iodine [6]. These data suggested an impaired bactericidal activity against isolates or strains with an elevated MIC value towards the biocidal agent.

Rose et al. described in 2009 that three out of four isolates of the *Burkholderia cepacia* complex with elevated MIC values for CHG (90–100 mg/L) could be equally killed in suspension tests by 4% CHG in 5 min compared to an ATCC control strain with a low MIC value (10–20 mg/L; Table 1). Only one isolate with an MIC value > 100 mg/L could not be sufficiently reduced by 4% CHG in 1 h [7].

Kawamura-Sato et al. categorized 20 nonrepetitive clinical isolates of *Acinetobacter* spp. with a low MIC value for CHG (10 mg/L) and 28 nonrepetitive clinical isolates of *Acinetobacter* spp. with a higher MIC value for CHG (>50 mg/L). The same classification was done for BAC (10 versus > 50 mg/L) and alkyldiaminoethylglycine hydrochloride (10–50 versus > 100 mg/L). A solution of 0.0032% CHG was able to equally reduce the bacterial load in 5 min by 5 $\log_{10}$, whereas concentrations of 0.0016% CHG and 0.0008% CHG demonstrated a significantly stronger bactericidal activity against the isolates with the low MIC value. Similar results were found with 0.0032% BAC and 0.0064% alkyl-diaminoethylglycine hydrochloride, which were effective against both types of isolates in 5 min but demonstrated a significantly stronger bactericidal activity at further dilutions to 50% and 25% against the isolates with the low MIC value (Table 1) [8].

Ciusa et al. selected triclosan-resistant mutants from *S. aureus* reference strains with MIC values of 4 mg/L compared to 0.12 mg/L for the reference strains. In suspension tests, the mean $\log_{10}$ reduction obtained with 0.01% triclosan in 5 min was comparable between all strains. With 0.06% triclosan, a larger $\log_{10}$ reduction was observed with the reference strain, whereas, with 0.1%, only one out of three mutant strains revealed a lower $\log_{10}$ reduction compared to the other strains (Table 1). The authors concluded that triclosan remained bactericidal against the mutant strains [9].

Thomas et al. found in six *P. aeruginosa* strains with increased MIC values to chlorhexidine diacetate similar $\log_{10}$ reductions by 0.01% chlorhexidine diacetate in suspension tests and carrier tests after 5 min compared to a control strain (Table 2), suggesting a comparable bactericidal activity despite elevated MIC values.

### 3.2. Can an Impaired Efficacy in Suspension Tests Predict an Impaired Efficacy under Practical Conditions?

#### 3.2.1. Bacteria

One study described both the bactericidal activity of various biocidal active substances in suspension tests (EN 13727) and carrier tests (EN 14561; Table 3). A carbapenem-resistant *K. pneumoniae* strain was isolated from duodenoscopes after an outbreak with 13 patients, despite strictly following the manufacturer's instruction for automated reprocessing using a commercially available product for disinfection based on 0.15% peracetic acid for 10 min suggesting that the outbreak strain may be tolerant to peracetic acid. The results obtained in suspension tests showed that a $\log_{10}$ reduction in 10 min required less than 0.0005% peracetic acid with the control strain, whereas the outbreak strain required a six-fold higher concentration of 0.003% peracetic acid to achieve the same effect. The susceptibility of both strains was similar against other biocidal active substances such as 5% hydrogen peroxide, 0.04% glutaraldehyde and 30% iso-propanol. In carrier tests, a concentration of 0.01% peracetic acid was necessary for a 5 $\log_{10}$ reduction in 10 min with the control strain, and the outbreak strain was reduced to the same extent with 0.15% peracetic acid [11]. The results suggest that an impaired bactericidal activity in suspension tests correlates with an impaired efficacy in carrier tests.

**Table 3.** Overview on the mean $\log_{10}$ reductions from suspension tests and carrier tests obtained with standard bacterial test strains and isolates with a suspected tolerance to peracetic acid.

| Species | Biocidal Active Substance | Strain | Mean Log$_{10}$ Reduction | | Exposure Time | Reference |
|---|---|---|---|---|---|---|
| | | | Suspension test | Carrier test | | |
| *K. pneumoniae* | 0.003% peracetic acid * | Outbreak strain (suspected tolerance to peracetic acid) | ≥5.0 | No data | 10 min | [11] |
| | ≤0.0005% peracetic acid * | ATCC 13883 | ≥5.0 | No data | 10 min | [11] |
| | 0.15% peracetic acid * | Outbreak strain (suspected tolerance to peracetic acid) | No data | ≥5.0 | 10 min | [11] |
| | 0.01% peracetic acid * | ATCC 13883 | No data | ≥5.0 | 10 min | [11] |
| | 5% hydrogen peroxide * | Outbreak strain (suspected tolerance to peracetic acid) | ≥5.0 | No data | 10 min | [11] |
| | | ATCC 13883 | ≥5.0 | No data | 10 min | [11] |
| | 0.04% glutaraldehyde * | Outbreak strain (suspected tolerance to peracetic acid) | ≥5.0 | No data | 10 min | [11] |
| | | ATCC 13883 | ≥5.0 | No data | 10 min | [11] |
| | 30% iso-propanol * | Outbreak strain (suspected tolerance to peracetic acid) | ≥5.0 | No data | 10 min | [11] |
| | | ATCC 13883 | ≥5.0 | No data | 10 min | [11] |

* Solution of biocidal active substance.

### 3.2.2. Mycobacteria

Three studies were identified with comparative data obtained in suspension tests and carrier tests with mycobacterial species suspected to be tolerant to glutaraldehyde (Table 4).

**Table 4.** Overview on the mean $\log_{10}$ reductions from suspension tests and carrier tests obtained with standard mycobacterial test strains and isolates with a suspected tolerance to biocidal active substances or disinfectants.

| Species | Biocidal Active Substance | Strain | Suspension Test | | Carrier Test | | Reference |
|---|---|---|---|---|---|---|---|
| | | | Mean log$_{10}$ reduction | Exposure time | Mean log$_{10}$ reduction | Exposure time | |
| *M. chelonae* | 2% glutaraldehyde * | NCTC 946 | >5.6 | 1 min | >5.3 | 1 min | [12] |
| *M. chelonae* | 2% glutaraldehyde * | Isolate from washer disinfector A (previous use of GDA) | 0.6–1.1 | 1 h | 0.0–0.2 | 1 h | [12] |
| *M. chelonae* | 2% glutaraldehyde * | Isolate from washer disinfector B (previous use of GDA) | 0.0–0.3 | 1 h | 0.0–0.2 | 1 h | [12] |
| *M. chelonae* | 1% active chlorine * | NCTC 946 | >5.1 | 1 min | >5.1 | 1 min | [12] |
| *M. chelonae* | 1% active chlorine * | Isolate from washer disinfector A (previous use of GDA) | >5.8 | 1 min | >5.0 | 1 min | [12] |
| *M. chelonae* | 1% active chlorine * | Isolate from washer disinfector B (previous use of GDA) | >5.2 | 1 min | >5.0 | 1 min | [12] |
| *M. chelonae* | 0.35% peracetic acid * | NCTC 946 | >5.5 | 4 min | >5.2 | 4 min | [12] |
| *M. chelonae* | 0.35% peracetic acid * | Isolate from washer disinfector A (previous use of GDA) | >5.1 | 1 min | >5.1 | 1 min | [12] |
| *M. chelonae* | 0.35% peracetic acid * | Isolate from washer disinfector B (previous use of GDA) | >6.1 | 1 min | >5.0 | 1 min | [12] |
| *M. chelonae* | 1.5% Glutaraldehyde* | ATCC 35,752 ("GDA susceptible") | >5.0 | 30 min | >5.0 | 45 min | [13] |
| *M. chelonae* | 1.5% Glutaraldehyde* | Strain 9917 ("GDA resistant") | 0.0 | 30 min | 2.3 | 45 min | [13] |
| *M. chelonae* | 1.5% Glutaraldehyde* | Strain Harefield ("GDA resistant") | 0.1 | 30 min | 3.6 | 45 min | [13] |
| *M. chelonae* | 1.5% Glutaraldehyde* | Strain Epping ("GDA resistant") | 0.0 | 30 min | 1.7 | 45 min | [13] |
| *M. abscessus* subsp. *massiliense* | 1.8% Glutaraldehyde * | CIP 108,297 ("GDA susceptible") | >5.0 | 30 min | >5.0 | 20 min | [13] |
| *M. chelonae* | 1.8% Glutaraldehyde * | ATCC 35,752 ("GDA susceptible") | >5.0 | 30 min | >5.0 | 20 min | [13] |

**Table 4.** *Cont.*

| Species | Biocidal Active Substance | Strain | Suspension Test | | Carrier Test | | Reference |
|---|---|---|---|---|---|---|---|
| *M. chelonae* | 1.8% Glutaraldehyde * | Strain 9917 ("GDA resistant") | 0.9 | 30 min | 2.3 | 20 min | [13] |
| *M. chelonae* | 1.8% Glutaraldehyde * | Strain Harefield ("GDA resistant") | 0.4 | 30 min | 4.4 | 20 min | [13] |
| *M. chelonae* | 1.8% Glutaraldehyde * | Strain Epping ("GDA resistant") | 1.0 | 30 min | 1.5 | 20 min | [13] |
| *M. abscessus* subsp. *abscessus* | 70% alcohol and 10% povidone iodine * | ATCC 19977 | 3.0 ** | 90 s | 4.6 *** <br> 2.0 **** | 2 min <br> 2 min | [14] |
| | 75% alcohol and 2% CHG * | ATCC 19977 | 0.0 ** | 90 s | 3.8 *** <br> 2.3 **** | 2 min <br> 2 min | [14] |
| *M. abscessus* subsp. *bolletii* | 70% alcohol and 10% povidone iodine * | BCRC 16915 | 5.4 ** | 90 s | 5.5 *** <br> 3.0 **** | 2 min <br> 2 min | [14] |
| | 75% alcohol and 2% CHG * | BCRC 16915 | 0.0 ** | 90 s | 5.3 *** <br> 2.6 **** | 2 min <br> 2 min | [14] |
| *M. abscessus* subsp. *massiliense* | 70% alcohol and 10% povidone iodine * | TPE 101 (outbreak strain) | 4.1 ** | 90 s | 1.2 *** <br> 0.9 **** | 2 min <br> 2 min | [14] |
| | 75% alcohol and 2% CHG * | TPE 101 (outbreak strain) | 0.0 ** | 90 s | 0.9 *** <br> 0.4 **** | 2 min <br> 2 min | [14] |

\* Commercially available formulation; ** products were only tested in dilutions in the suspension tests, data for the 1:25 dilution (smallest dilution) are presented; *** clean conditions; **** dirty conditions.

In 1997, Griffiths et al. described two *M. chelonae* strains that were consistently isolated from two separate washer disinfectors and from reprocessed endoscopes. Glutaraldehyde was used for the disinfection step. Both isolates were only marginally reduced by 2% glutaraldehyde in suspension tests within 1 h (0.0–1.1 $\log_{10}$), whereas a control strain was strongly reduced in 1 min (>5.6 $\log_{10}$). Similar results were found in carrier tests. No relevant difference of mycobactericidal efficacy was found between the outbreak strains and the control strain when 1% active chlorine or 0.35% peracetic acid was used (Table 4) [12].

Burgess et al. evaluated the mycobactericidal activity of 1.5% and 1.8% glutaraldehyde against the same glutaraldehyde-resistant *M. chelonae* strains described by Griffiths et al. in comparison to two glutaraldehyde-susceptible mycobacterial control strains. Solutions of 1.5% and 1.8% glutaraldehyde had poor mycobactericidal activity in 30 min against the outbreak strains in the suspension test (0.0–0.9 $\log_{10}$), whereas the mycobactericidal activity was much better in 30 min against the control strains (>5.0 $\log_{10}$). Similar results were found in the carrier tests. In the carrier tests, the solution of 1.5% glutaraldehyde had some mycobactericidal activity in 45 min against the outbreak strains (2.3–3.6 $\log_{10}$) and sufficient mycobactericidal activity against the control strains (>5.0 $\log_{10}$). Similar results were found with 1.8% glutaraldehyde (Table 4) [13].

Cheng et al. investigated the mycobactericidal activity of two skin antiseptics based on a combination of 70% "alcohol" plus 10% povidone iodine and 75% "alcohol" plus 2% CHG after an outbreak of skin and soft tissue infections caused by *M. abscessus* subsp. *massiliense*. Two control strains were included. The suspension tests were only done with dilutions of the skin antiseptics, with 1:25 being the weakest dilution, resulting in concentrations of biocidal active substances that were far below use concentrations. The product containing povidone iodine showed some mycobactericidal activity in 90 s against all three strains (3.0–5.4 $\log_{10}$), whereas the product containing chlorhexidine had no mycobactericidal activity (0.0 $\log_{10}$) when diluted by 1:25. Undiluted products were applied in the carrier tests for 2 min. With the outbreak strain, only a poor mycobactericidal efficacy was found with both skin antiseptics (0.4–1.2 $\log_{10}$); the efficacy was better with the control strains (2.0–5.5 $\log_{10}$). The presence of an organic load impaired the efficacy against all strains (Table 4) [14].

## 4. Discussion

MIC values are quite easy to obtain, although some variations have been described [15]. However, in three of the five studies, the elevated MIC values did not correlate with an impaired bactericidal efficacy measured with biocidal active substances or disinfectants against the biocide-tolerant isolates. In addition, biocides are often used at concentrations

exceeding the MIC value. Biocidal active substances are also typically used as part of formulations that may be based on two or more active substances and whose additional ingredients will impact on the overall antimicrobial efficacy. That is why the MIC values may only be used as a trend indicator [16].

Suspension tests and carrier tests require more time and laboratory materials compared to MIC tests. In one of two studies with bacteria, an impaired bactericidal activity in the suspension tests was described that correlated with an impaired efficacy in the carrier tests. In the other study, the results from the suspension tests did not indicate an impaired bactericidal activity, which was confirmed by comparable results for the mutant and control strains in the carrier tests. In two of three studies with mycobacteria, an impaired mycobactericidal activity in the suspension tests was described that correlated with an impaired efficacy in the carrier tests. In the remaining study, the suspension tests were not carried out with the undiluted skin antiseptics, so that a comparison to the results obtained with the carrier tests could not be made. Based on the available evidence, there seems to be a good correlation between an impaired microbiocidal activity in the suspension tests and an impaired efficacy in the carrier tests. However, the results from the suspension tests still have a relevant limitation. In 2018, "iso-propanol-tolerant" *E. faecium* isolates were described. An evaluation of 139 isolates collected between 1997 and 2015 revealed that isolates after 2010 were 10-fold more tolerant to killing by 23% iso-propanol in suspension tests than older isolates [17]. For surface and hand disinfection, however, the results had no practical relevance, because the commonly used concentrations of 60% and 70% iso-propanol were equally effective against "iso-propanol-tolerant" and control strains [18]. That is why the concentration of the biocidal active substance or disinfectant used in the suspension tests should be the one used in clinical practice in order to obtain clinically relevant data.

Definitions to describe the resistance to biocidal active substances and disinfectants are currently not harmonized internationally. Russell proposed in 2003 to classify an isolate as resistant that is not inactivated by an in-use concentration of a biocide or a biocide concentration that inactivates other strains of that organism [19]. Maillard proposed in 2018 to regard an isolate as resistant that is surviving in a biocidal product [20]. That is why there is a need to have a consensus definition for resistance, epidemiological cut-off values and clinical resistance breakpoints for biocidal active substances and disinfectants [21].

For antibiotics, resistance is the ability of bacteria to replicate and not just survive in the presence of an antibiotic. It is typically caused by inherited mutations and associated with numerous molecular mechanisms such as the horizontal gene transfer of resistance-encoding genes encoding for antibiotic inactivating enzymes or efflux pumps [1,2]. As a result, a higher concentration of the antibiotic is required to produce the same effect in a resistant strain as is produced in a susceptible strain [3]. Tolerance, however, is the ability of a bacterial population to survive a transient exposure to antibiotics, even at concentrations that far exceed the MIC. Two main forms of tolerance have been identified: "tolerance by slow growth", which occurs at steady state, and "tolerance by lag", which is a transient state that is induced by starvation or stress [3].

Comparable findings regarding resistance were described with biocidal active substances, such as the replication of isolates in solutions of biocidal active substances or disinfectants. For example, isolates of *Achromobacter* spp. [3] were detected from contaminated surface disinfectant tissue dispensers containing products based on surface-active biocidal agents with a proven bactericidal activity within 1 h [22]. In follow-up experiments, it was found that the isolates were able to multiply in the use solution of the disinfectants at room temperature within 1–4 weeks, resulting mostly in a bacterial load of $10^7$ CFU per ml [23]. Another report described various Gram-negative species that were isolated from 28 in-use bottles of liquid soap based on 2% CHG, including a pan-resistant *A. baumannii*, a multi-resistant *P. aeruginosa* and a pan-resistant *K. pneumoniae*. All three isolates were able to multiply in a 1:1 dilution of the soap in tryptic soy broth containing 1% CHG within 24 h [24]. In another report, *S. marcescens* was isolated from 2% CHG solutions in five

plastic containers at a concentration of $10^8$ CFU per mL. The authors thought that the contamination probably occurred in the pharmacy when the plastic stock bottles were filled. CHG was supplied by the manufacturer as a 4% solution and diluted locally with tap water to 2%. Empty bottles were returned from the wards to the pharmacy, rinsed with tap water and refilled [25]. The high bacterial load in the plastic containers is very likely explained by a multiplication of the isolates in the 2% CHG solution. Increased rates of mutation have been described with *E. coli* after exposure to CHG, DDAC or copper, and a higher of conjugation were found after exposure to BAC or CHG [26]. CHG and triclosan were also able to increase the horizontal gene transfer of antibiotic resistance genes to a recipient *E. coli* strain [27]. Other reports described resistance-encoding genes encoding for biocide inactivating enzymes, efflux pumps and physiological and metabolic cellular changes [20]. These examples show that the criteria for resistance defined for antibiotics may also be fulfilled with biocidal active substances or disinfectants.

With antibiotics, the additional term "persistence" has been defined. It is the ability of a subset of the population to survive exposure to a bactericidal drug concentration. The hallmark of persistence is the biphasic killing curve showing that not all bacterial cells in a clonal culture are killed at the same rate. When persister cells regrow in the absence of an antibiotic, their progeny give rise to a population that is as susceptible to the antibiotic as the parental population it was isolated from. The size of the persister subpopulation will only weakly depend on the concentration of the antibiotic, as long as it is far above the MIC. In contrast to resistant cells, persisting bacteria cannot replicate in the presence of the antibiotic any better than the non-persister cells but are killed at a lower rate than the susceptible population from which they arose. This property also distinguishes persistence from heteroresistance, a phenomenon in which a small subpopulation transiently displays a substantially (more than eight-fold) higher MIC. In the context of disinfectants, persisters may be found in the biofilm, where they are typically less susceptible to the biocidal agent or disinfectant [10,28]. The deeper layers of a biofilm contain more starving cells, dormant cells, VBNC cells and persisters, which may all be difficult to reach by the biocidal agent or disinfectant [29].

In analogy to antibiotics, a suitable definition for resistance against biocidal active substances or disinfectants may be the ability of bacteria to replicate and not just survive in the presence of a biocidal agent or disinfectant. It is typically caused by inherited mutations and associated with numerous molecular mechanisms, such as the horizontal gene transfer of resistance-encoding genes encoding for antibiotic inactivating enzymes or efflux pumps. As a result, a higher concentration or a longer exposure time of the biocidal agent or disinfectant is required to produce the same effect in a resistant strain as is produced in a susceptible strain.

A limitation of the findings is that only few studies were found, so upcoming research may yield different results. Another limitation is the lack of data with clinically relevant fungi or those detected in food processing and production. Future research will hopefully cover these aspects and may provide a better understanding of which of the methods (MIC, MBC, suspension tests and tests under practical conditions) provides the most reliable predictive value for the identification of biocide-resistant isolates.

## 5. Conclusions

The determination of the MIC values is suitable to determine the susceptibility trends of species, e.g., in a historical context or under frequent exposure to biocidal active substances and disinfectants. The results from the suspension tests and tests under practical conditions such as carrier tests allow to determine a clinically relevant resistance.

**Funding:** This research received no external funding.

**Institutional Review Board Statement:** Not applicable.

**Informed Consent Statement:** Not applicable.

**Data Availability Statement:** Not applicable.

**Conflicts of Interest:** The author has received honoraria from Schülke & Mayr, Germany, outside the submitted work. The views expressed here are those of the author and do not necessarily reflect those of the university he is affiliated with.

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
