# Peer review of "Suitability of Methods to Determine Resistance to Biocidal Active Substances and Disinfectants—A Systematic Review"

_2673-947X, doi:10.3390/hygiene2030009_

Round 1

Reviewer 1 Report

This is an outstanding research in the field of microbial resistance and widely used disinfectants. The literature was appropriate and tables and figures are well presented. There is no need to improve body text, except I would appreciate if the authors would add limitation section.

Author Response

This is an outstanding research in the field of microbial resistance and widely used disinfectants. The literature was appropriate and tables and figures are well presented. There is no need to improve body text, except I would appreciate if the authors would add limitation section.

Many thanks for your feedback. I have added two limitations at the end of the discussion.

Reviewer 2 Report

General Comment

1. A noteworthy contribution, that has not been made previously.

2. Comparison of MIC measurements with "practical" suspension or carrier susceptibilities have not been made, but simply assumed to agree. However, as this brief review shows, in 60 % of instances, the MIC results are not predictive of suspension or carrier results.

Specific Comments

Abstract

Lines 7-24. Please consider shortening the section and using more direct language. Sentences at the end (Lines 20-24) can be deleted and emphasis placed on the sentence in lines 18-20 should be emphasized.

Background

Lines 29-57. Please consider deleting in order to get to the point of the information immediately.

Lines 76-87. Well-written and presented.

Results

Lines 147-148. The word "practical" to describe suspension or carrier tests is new and its inclusion here might confuse the reader. Please use a single standard terminology. From one perspective, MIC and suspension or carrier tests are all "practical".

Tables 1-4. Well presented tabulations, but is there a numerical or statistical method for comparison of the data from the different methods; for example data for MIC mg/l values with data for suspension mean log reductions. Perhaps our author has realized that that lack of method may be one reason for the fact that no prior comparisons have been made. Any suggestions? I wonder whether MBC values could be used, as they identify the concentration of agent resulting in killing 99.9% of the initial inoculum and therefore can be compared to bactericidal activity in suspension.

Discussion

Lines 238-257. Please consider adding this to the Introduction.

Lines 258-269. This section, as well as that in the Introduction (lines 49-57) should be re-directed to considering the problem of comparing MIC data with  that from suspension or carrier-killing. 

Lines 270-294. This is evidence of a detective story involving identification of a source of infection/contamination based on antimicrobial resistance. Interesting, but not integrated into the focus of the manuscript.

Lines 295-318. One way to overcome the absence of definitions would be to develop methods to compare MIC/MBC data with killing in suspension or on carriers.

Conclusions

Lines 320-326. Unnecessary and the final sentence, while thoughtful, is speculation.   

Author Response

General Comment

  1. A noteworthy contribution, that has not been made previously.
  2. Comparison of MIC measurements with "practical" suspension or carrier susceptibilities have not been made, but simply assumed to agree. However, as this brief review shows, in 60 % of instances, the MIC results are not predictive of suspension or carrier results.

Many thanks for your feedback.

Specific Comments

Abstract

Lines 7-24. Please consider shortening the section and using more direct language. Sentences at the end (Lines 20-24) can be deleted and emphasis placed on the sentence in lines 18-20 should be emphasized.

The abstract has been shortened, the content in lines 18-20 has been emphasized.

Background

Lines 29-57. Please consider deleting in order to get to the point of the information immediately.

Most of the lines have been deleted, a small part has been moved to the discussion.

Lines 76-87. Well-written and presented.

Thank you.

Results

Lines 147-148. The word "practical" to describe suspension or carrier tests is new and its inclusion here might confuse the reader. Please use a single standard terminology. From one perspective, MIC and suspension or carrier tests are all "practical".

The word “practical” is used for “tests under practical conditions” simulating the practical use of disinfectants. Its definition is described in the introduction in lines 76-78 and is used as such in the framework of European Norms for testing disinfectants. It does not include suspension tests. I hope that my explanation helps to understand the use of this word.

Tables 1-4. Well presented tabulations, but is there a numerical or statistical method for comparison of the data from the different methods; for example data for MIC mg/l values with data for suspension mean log reductions. Perhaps our author has realized that that lack of method may be one reason for the fact that no prior comparisons have been made. Any suggestions? I wonder whether MBC values could be used, as they identify the concentration of agent resulting in killing 99.9% of the initial inoculum and therefore can be compared to bactericidal activity in suspension.

Excellent comment, many thanks. I am not aware of any numerical or statistical method for a comparison of data obtained with different methods. The current test system within the European Norms only allows to describe the extent of antimicrobial activity as sufficient (e.g. at least 5 log10 reduction against bacteria) or insufficient (e.g. < 5 log10 reduction against bacteria). I have doubts if the MBC values will help in that respect but it may be a future research project to find out if the increased MBC value has a reliable predictive value to identify a biocide-resistant isolate. This aspect is integrated in the last sentence of the discussion.

Discussion

Lines 238-257. Please consider adding this to the Introduction.

I prefer to leave it in the discussion because it is a brief description of the results of the review plus an example describing the limitation of results obtained in suspension tests. It may not be a good place to read the results of the review in the introduction.

Lines 258-269. This section, as well as that in the Introduction (lines 49-57) should be re-directed to considering the problem of comparing MIC data with  that from suspension or carrier-killing. 

I am afraid that I do not really understand this comment. Lines 258-269 describe the definitions for resistance and tolerance for antibiotics and are not directly related to the problem of comparing MIC data with that from tests under practical conditions. Lines 49-57 were proposed by the reviewer to be deleted (see above under Background) where I described that a part of it has been moved to the discussion. That is why I have not made any changes based on this comment, only the one that has been described before for the Background Section which takes into account the problem described by the reviewer.

Lines 270-294. This is evidence of a detective story involving identification of a source of infection/contamination based on antimicrobial resistance. Interesting, but not integrated into the focus of the manuscript.

I respectfully disagree. In line 257 I describe the definition for resistance for antibiotics as the ability of bacteria to replicate and not just survive in the presence of an antibiotic. In lines 270-294 I describe examples of bacterial multiplication in the presence of disinfectants so that the definition of resistance for antibiotics may also be used with disinfectants. In that respect I consider this paragraph to be within the focus of the manuscript.

Lines 295-318. One way to overcome the absence of definitions would be to develop methods to compare MIC/MBC data with killing in suspension or on carriers.

Yes, that may be true, I have added this aspect at the end of the discussion.

Conclusions

Lines 320-326. Unnecessary and the final sentence, while thoughtful, is speculation. 

Lines 320-326 have been deleted. I have also deleted the final sentence because it is not directly related to the question raised in the article although the study by Brunke et al. provides evidence for the statement so that I do not consider it to be speculation.